# Suitability of High-Resolution Mass Spectrometry for Routine Analysis of Small Molecules in Food, Feed and Water for Safety and Authenticity Purposes: A Review

**DOI:** 10.3390/foods10030601

**Published:** 2021-03-12

**Authors:** Maxime Gavage, Philippe Delahaut, Nathalie Gillard

**Affiliations:** CER Groupe, Rue du Point du Jour 8, 6900 Marloie, Belgium; m.gavage@cergroupe.be (M.G.); n.gillard@cergroupe.be (N.G.)

**Keywords:** routine testing, high-resolution mass spectrometry, food, feed, water, veterinary drug residues, natural toxins, pesticides, food authenticity

## Abstract

During the last decade, food, feed and environmental analysis using high-resolution mass spectrometry became increasingly popular. Recent accessibility and technological improvements of this system make it a potential tool for routine laboratory work. However, this kind of instrument is still often considered a research tool. The wide range of potential contaminants and residues that must be monitored, including pesticides, veterinary drugs and natural toxins, is steadily increasing. Thanks to full-scan analysis and the theoretically unlimited number of compounds that can be screened in a single analysis, high-resolution mass spectrometry is particularly well-suited for food, feed and water analysis. This review aims, through a series of relevant selected studies and developed methods dedicated to the different classes of contaminants and residues, to demonstrate that high-resolution mass spectrometry can reach detection levels in compliance with current legislation and is a versatile and appropriate tool for routine testing.

## 1. Introduction

Food and feed analysis is essential to guaranty their quality, authenticity and safety. Analytical strategies have been developed for decades to evaluate food and feed composition and nutritional value and to detect the presence of undesirable or harmful compounds or foodborne pathogens. The analysis of chemical substances in food and feed is a challenging task, given the multitude of matrices encountered and the disparate properties of targeted contaminants [1]. Moreover, some of these substances must be detected and/or quantified at trace levels with sufficient accuracy and robustness.

In Europe, a high level of consumer protection is required by Article 152 of the Treaty establishing the European Community [2]. To reach this high level of health protection, a risk analysis procedure based on scientific evaluation and including factors such as the feasibility of control underpins Community legislation. The aim is to establish the optimal balance between the risks and benefits of substances that are used intentionally, focusing on the reduction of contaminants. The legislation separately considers different classes of chemical substances, including contaminants and residues. The legislation on contaminants is based on scientific advice and the principle that contaminant levels should be kept as low as can be reasonably achieved following good working practices. Maximum levels have been set for certain contaminants (e.g., mycotoxins, dioxins, heavy metals, nitrates and chloropropanols) in order to protect public health [3]. The legislation on residues of veterinary medicinal products used in food-producing animals and on residues of plant protection products (pesticides) provides for scientific evaluation before respective products are authorised. If necessary, maximum residue limits (MRLs) are established, and in some cases, the use of substances is prohibited [4,5]. This present review focus on the European legislation. Complementary information on the regulation and safety assessment of food substances in various countries and jurisdictions can be found in the review of Magnuson and co-workers [6].

Water is one of the most important resources, and its preservation is a major challenge, regarding both the environment and humans. Synthetic pesticides used intensively in agriculture can enter surface waters, mainly due to runoff driven by precipitation or irrigation. Pharmaceuticals for both human and veterinary purposes are excreted, and the unaltered parent compounds or their metabolites and can be deposited in environmental waters as a consequence of incomplete elimination by wastewater-treatment plants. However, efforts are being undertaken to develop new systems to degrade pharmaceutical products in wastewater treatment plants [7]. This mixture of chemicals, pharmacologically active compounds and their transformation products are potentially harmful to aquatic life and humans when they enter drinking water. The justified concern over this hazard has led to the development of analytical methods for measuring freshwater contamination.

For years, mass spectrometry has been considered the most suitable analytical technique for the detection of multiple compounds in food, feed and water. Coupled to liquid chromatography (LC), high-performance LC and ultra-high performance LC (HPLC, UHPLC) or gas chromatographic (GC) separation with an ionization source such as electrospray (ESI), a large number of mass spectrometry-based methods were developed to comply with updated regulations. Most of the developed methods use triple-quadrupole instruments, and the best-performing of these are able to sensitively and accurately detect and quantify more than 1000 compounds in a single analysis [8,9,10,11]. These instruments are defined as low-resolution mass spectrometers, with a typical resolution of approximately 1 atomic mass unit for quadrupole analysers [12]. The combination of chromatographic separation and the use of multiple reaction monitoring (MRM) mode, working like a noise-reducing double filter, allow enhanced sensitivity and selectivity. However, the use of triple quadrupole instruments has proven to have some limitations, such as a limited number of compounds targeted during the analysis. The sensitivity of MRM methods strongly depends on the length of the chosen dwell-time. Therefore, the more transitions to be monitored, the shorter the resulting dwell-time and the poorer the obtained sensitivity [13]. This sensitivity issue of multiple-compounds methods can be balanced by the use of retention time-based MRM windows. In addition, the use of triple quadrupole instruments and MRM methods is limited to targeted analysis. To effectively apply this approach, the structure of the compound must be characterized before its detection. Methods development can be time-consuming, and standards must be acquired to optimize compound-specific instrumental conditions, including transition selections, ion-source voltages, and collision energies [14]. MRM methods are, therefore, unable to screen for unknown compounds.

In the last decade, high-resolution mass spectrometry (HRMS) has become more accessible, particularly with the development of Orbitrap MS-based instruments and the improvements to time-of-flight (ToF) MS systems. As for triple quadrupole instruments, high-resolution mass spectrometers can be coupled to chromatographic separation units. Orbitrap and ToF systems are versatile instruments with fast scan velocities, sufficient dynamic ranges and the possibility of tandem mass spectrometry (MS/MS) when used as components of hybrid instruments (combining a quadrupole analyser with Orbitrap (Q-Orbitrap) and ToF (Q-ToF)). HRMS instruments are usually described as less sensitive than triple quadrupoles [15]. However, thanks to recent technical improvements such as the introduction of new ion transition devices or advances in detection technology [1], several studies presented similar sensitivities achieved by the two types of instrument [16,17,18]. Moreover, a higher resolution provides an enhanced selectivity when a large number of analytes are determined simultaneously and, for the best-performing instruments with sufficient resolving power, the potential to discriminate analytes from isobaric co-eluting sample matrix compounds. The use of ion mobility-coupled chromatographic separation and HRMS is particularly powerful for this purpose [19]. Complementary technical information on HRMS instruments including separation techniques, ionization and acquisition modes can be found in the review from Yan and co-workers [20].

The major advantage of HRMS systems is the ability to record a theoretically unlimited number of compounds in full-scan mode with additional structural information, using hybrid instruments. Acquired data can be processed using target analysis, suspect screening and non-target screening. Moreover, the high volume of stored full-scan or MS/MS data can be retrospectively analysed without sample re-injection. Finally, untargeted HRMS analysis can be combined with multivariate chemometric tools to efficiently extract relevant information from very complex datasets. This statistical analysis helps in the exploration of specific biomarkers that can categorize/differentiate the analysed samples. The chemical profiling of samples could be focused on the m/z values, which vary significantly from one sample category to another [21].

Thanks to these strengths and the ever-increasing number of new analytes that must be monitored, HRMS analyses are increasingly accepted for multi-residue analysis [15]. This type of instrument is, however, still rarely used for routine analysis by control laboratories. For instance, in 2019, only 4 of the 26 participants in 19 PU (study code) proficiency testing for the screening of antibiotic residues in pork muscle used HRMS. Additionally, in the 2019 Fapas^®^ food chemistry proficiency test for the detection of avermectins and anthelmintics in bovine liver, only 3 out of the 32 participants used HRMS [22]. The proportion of laboratories using HRMS is higher for proficiency tests for pesticides in fruits and vegetables. For instance, 33% of the laboratories participating in EUPT-FV-SM11 [23], pesticide residues in red cabbage homogenate (with 67 participants in 2019), and EUPT-FV-SM10 [24], pesticide residues in green bean homogenate (with 69 participants in 2018), used HRMS.

To demonstrate the applicability of HRMS in routine analyses, several relevant examples were collected over the last decade. The selected studies are presented according to the type of targeted compounds, pesticides, veterinary drug residues and toxins. Given the high number of compounds that can be detected in single analysis, a section is dedicated to multi-class analysis. The analysis of water constitutes a significant portion of the HRMS-related literature. The selected studies were, however, limited to drinking water to maintain a focus on the ‘food and feed’ topic, excluding the environmental aspects of water. Finally, quite apart regulated compounds detection, the potential of HRMS-based analysis combined with chemometrics tools for food authenticity control is addressed.

## 2. Analysis of Pesticides

Synthetic pesticides play a major role in food and feed production. Their use has helped to immensely increase agricultural productivity and resist the ever-increasing demographical and economical pressure. Pesticides are used to protect crops, including fruits, vegetables, cereals and fibre plants, against insects, plant pathogens, weed and fungi. It has been estimated that nearly one-third of all agricultural products are produced using pesticides, and without them, the loss of fruits, vegetables and cereals from pest injury could increase to 78%, 54% and 32%, respectively [25]. However, the use of pesticides is associated with negative external effects, e.g., pollution of waterways and non-target ecosystems, risks for human health and costs for monitoring of residues on food.

In 2003, the work of Klein and Alder [26] was considered a masterpiece in the field of pesticide analysis. The LC-MS/MS method they developed, based on a triple quadrupole instrument, was able to screen and quantify 100 pesticides and metabolites in various crops. Nowadays, it is estimated that almost 1000 different pesticides could potentially be used in agriculture [14]. The main challenge is the ability to economically analyse the presence of these chemicals, their metabolites, and degradation products when precise knowledge of pesticide application or misuse is lacking.

Given the large number of compounds to detect, HRMS is particularly well-suited for this purpose. Zhibin Wang and co-workers [27] developed a qualitative screening and identification strategy for 317 pesticides in fruits and vegetables using LC-Q-ToF. The strategy is based on two injections of each sample extract. In the first chromatographic run, the single full-scan MS mode was performed and the sample was screened for possible target compounds. Potential contaminants were confirmed in a second chromatographic run under targeted MS/MS conditions in which the resulting product ion spectra were used to search a homemade MS/MS library. In studies from Jian Wang and co-workers, identification and quantification of pesticides were performed using the same approach. A sequential combination of a full MS scan for quantification and a data-dependent MS/MS scan for confirmation was used for the analysis of 166 pesticides in fruits and vegetables [28]. The method was later extended to 451 pesticides residues and validated via an evaluation of overall recovery, intermediate precision, limits of detection (LOD) and quantification (LOQ) and measurement uncertainty [29]. For the 10 studied matrices, 94.5% of the pesticides in fruits and 90.7% of those in vegetables had recoveries of between 81% and 110%; 99.3% of the pesticides in fruits and 99.1% of those in vegetables had an intermediate precision ≤20%; and 97.8% of the pesticides in fruits and 96.4% of those in vegetables showed a measurement uncertainty ≤50%.

In another study from Gómez-Ramos and co-workers [30], the authors used LC-Q-Orbitrap MS for the analysis of pesticide residues in fruit and vegetable commodities. The system was used to detect, identify and quantify, in various extracts (tomato, pepper, orange and green tea), 139 pesticides, all of which were included in the European Union Monitoring Program. Here, detection, identification and quantification were achieved within the same analysis, combining full scan data acquisition for detection and quantification and MS/MS data for identification. Extracts spiked with a mixture of the analysed pesticides at 10, 50, 100 or 500 μg/kg were analysed using both LC-Q-Orbitrap MS and a triple quadrupole instrument. A comparison of the results showed that these two systems have similar capabilities for quantification, with the advantage of a better selectivity for HRMS as well as the possibility to perform retrospective analysis. In a recent study by Kiefer and co-workers [31], the authors performed a suspect screening for over 300 pesticides and over 1100 pesticide transformation products in 31 Swiss groundwater samples. This study aimed to comprehensively assess the impact of agricultural pesticide application on groundwater quality. Suspect screening was combined with HRMS analysis to overcome the lack of reference material for most of the transformation products. The acquired data were used to search the suspect list containing the monoisotopic masses of expected compounds. The suspect hits were then checked for plausibility, with criteria including background interference, retention time, isotope pattern, ionization potential and MS/MS fragmentation. The authors demonstrated the importance of considering transformation products in analysis, with the total concentration of pesticide transformation products exceeding the total concentration of the active substances in 30 samples. One of the findings of the study was that the concentration of 15 transformation products of 9 pesticides exceeded 100 ng/L in at least one sample, demonstrating the importance of such an analysis.

In previous studies, reverse-phase liquid chromatography was used to separate the analytes before HRMS analysis. However, highly polar pesticides have poor retention with this type of column and are co-eluted with unwanted co-extractive substances. To successfully analyse highly polar pesticides and avoid derivatization steps or single-residue analysis, Gasparini and co-workers [32] developed a method based on HRMS and ion chromatography as a separating technique. The method was validated for the quantification of 11 highly polar molecules (four pesticides and relative metabolites) in fruit, cereals and honey. Several proficiency tests, used to verify procedure performance, demonstrated that the method is fit for the purpose of routine analysis in an official laboratory.

Gas chromatography combined with mass spectrometry was traditionally used for pesticide analysis. However, the use of this separation technique requires that the analytes are volatile and thermally stable. To extend analysis applicability to a wider range of compounds, without the need of prior derivatization, strategies gradually changed to liquid chromatography with similar performance [33]. Nevertheless, the properties of several pesticides are not compatible with LC separation, and GC has been required and used in some recent publications. The use of GC-Q-ToF, combining full scan with MS/MS experiments and using accurate mass analysis, was explored by Besil and co-workers [34] for the automated determination of 70 pesticide residues in fruits and vegetables. In addition to satisfying validation results at low targeted contamination levels (1, 5 and 10 µg/kg), the authors pointed out the limited dynamic range of the method as a potential limitation for quantification. However, this problem can be overcome by selecting characteristic ion fragments with lower abundance or sample dilution, which necessitate a second analysis. Vargas-Pérez and co-workers [35] proposed an application combining targeted and non-targeted approaches, using GC-Q-Orbitrap, for the multi-residue analysis of multiple pesticides in fruits and vegetables. The targeted method was successfully validated for 191 pesticides. When applied to real unknown samples, targeted and untargeted methods generated the same results. In addition, data acquired with the untargeted method were compared with a library containing more than 200,000 spectra (containing multiple classes of compounds, such as metabolites, drugs, small peptides, lipids or glycans, in addition to pesticides). Results were based on a search index score indicating the match quality between the library hit and deconvolved experimental spectrum.

As demonstrated by the numerous studies presented in this section and summarized in Table 1, HRMS became a key element in the analysis of pesticides. In the near future, thanks to the combination of separation methods coupled to HRMS, databases and software tools, it is estimated that methods capable of screening up to 1000 pesticides and metabolites will be achievable [14].

## 3. Analysis of Veterinary Drug Residues

The use of veterinary drugs or veterinary medicinal products (such as antibiotics, antiprotozoals, anthelmintics, anti-inflammatory, corticosteroids or hormones substitutes) has become essential to providing a sufficient amount of food for the growing world population. For the purpose of increasing productivity, drugs improve the rate of weight gain, improve feed efficiency or prevent and treat diseases in food-producing animals [36]. However, the use of veterinary drugs is associated with health hazards for the consumer of animal food products, including meat, fat, milk, egg, fish, seafood, honey and derived products. The presence of veterinary drug residues in food might induce various effects, such as allergic reactions, carcinogenic or teratogenic mechanisms or antimicrobial resistance [37].

On the basis of the scientific assessment of the safety of those substances and to protect public health, the presence of veterinary drug residues in foodstuffs of animal origin is regulated by the European Union (Commission Regulation EC No 2377/90) [38] with imposed maximum residue limits (MRLs). Based on the lowest acceptable daily intake, together with metabolism and residue depletion studies, the MRLs of the residues are determined for each tissue, expressed in micrograms per kilogram on a fresh-weight basis [39]. In most of the cases, the MRLs are related to the parent compounds, but they could also be based on single metabolites or a mixture of compounds. Some substances, such as chloramphenicol or nitrofurans, are totally prohibited and MRLs are not established.

Pharmacokinetic parameters of administered veterinary drugs are extensively studied to establish a withdrawal period. This period between last administration and slaughter ensures that food from treated animals will not exceed the MRL and can be eaten safely by humans. This withdrawal period is drug related with specific absorption and elimination rates and also depends on the route of administration and the dosage regimen [40]. However, besides illegal use, a series of causes, such as producer mistakes, disease state or concomitant use of different drugs, may lead to failure to comply with MRLs. Therefore, veterinary drug residue analysis is required to verify compliance with MRLs and detect the presence of prohibited substances. Today, there are approximately 200 veterinary drug residues that must be controlled in foodstuffs [41].

Most of the current methods used for the analysis of multiple veterinary drugs residues are based on liquid chromatography coupled to tandem mass spectrometry, using a triple quadrupole mass analyser programmed to acquire data for selected ion transitions corresponding to the analytes of interest. In recent years, HRMS-based methods were proposed with the advantage of potentially analysing an unlimited number of compounds using a full-scan acquisition mode. This acquisition mode is also well-suited to monitoring drug metabolites that could be more stable and/or toxic than parent drugs but are seldom commercially available as reference substances [42]. In this case, the optimization of analyte-specific MS/MS transitions with triple quadrupole instruments is more complicated.

Nearly a decade ago, Romero-González and co-workers [43] demonstrated the potential of HRMS for the determination of veterinary drugs in milk and its applicability for routine analysis. They compared three methods (running time <4 min) based on Orbitrap, quadrupole time of flight and triple quadrupole instruments for the screening of 29 veterinary drugs from different classes. Overall better results, in terms of cut-off values and uncertainty regions, were obtained using the Orbitrap-based screening method. Additionally, the Orbitrap method showed good quantitative results for all the studied analytes, and the limits of quantification were, except for one compound, lower than the MRLs established for the European Union (EU). Berendsen and co-workers [44] organised an inter-laboratory study including 21 laboratories to evaluate the use of different low- and high-resolution MS techniques and acquisition modes, with respect to the selectivity of 100 veterinary drugs in liver tissue, muscle and urine. For complex matrices, they concluded that only targeted MS/MS monitoring a single product ion in HRMS using a (with a maximum of 5 ppm mass deviation), yields comparable selectivity and false positive and negative rates as triple quadrupole monitoring two product ions. Authors highlighted the clear advantage that data acquired with HRMS instruments can be retrospectively analysed. Another method, based on an Orbitrap analyser, was proposed by León and co-workers [45] for the multi-residue screening of 87 banned or unregulated veterinary drugs in urine. The method was validated, and the detection capability (CCβ) established levels were equal to or lower than the recommended concentrations established by EU reference laboratories. The authors concluded that HRMS is a powerful and reliable tool for the identification of substances in multi-class multi-residue analysis and could be used on a routine basis in the official food safety laboratories.

In more recent studies and published methods, the number of screened drugs increased, as did the number of matrices considered in validation. Staub Spörri and co-workers [46] developed, for instance, a method based on a time-of-flight instrument for the screening of 200 veterinary drugs in honey. Boix and co-workers [47] developed another screening method based on quadrupole time-of-flight mass spectrometry and covering 116 human and veterinary drugs. The method was validated in five types of animal feed at 0.02 and 0.2 mg per kg. The method was successfully applied to real feed samples, and two hormones banned in Europe were detected in some samples. The authors also evaluated the applicability of their screening method to quantitative analysis. Quantification was performed using calibration standards in solvents and relative responses to isotope-labelled internal standards (ILIS) for matrix effects correction. They concluded that, due to the strong matrix effects resulting from the matrix complexity and little sample manipulation, the analyte-labelled ILIS was required to ensure an adequate quantification. Kaklamanos and co-workers [48] considered another approach for the quantification of 48 antimicrobial agents from a wide range of chemical groups/families. Using an Orbitrap instrument, the target analytes were quantified using the standard addition approach. The authors argued that this approach is safer and easier, given the high complexity of animal feed and the lack of blank representative material. The use of a one-point standard addition was shown to be suitable for the accurate determination of the analytes and reduction of the workload, which constitutes a main advantage for using the method in routine analysis. Alcántara-Durán and co-workers [49] developed a multi-residue method, based on an Orbitrap instrument, for the analysis of 87 veterinary drugs in honey, veal muscle, egg and milk. By optimizing the parameters of the method and applying a dilution factor of 100 to the sample, matrix effects were completely removed for all the compounds and matrices tested. Sensitivity decrease due to sample dilution was balanced by downscaling the size of liquid separations using nanoflow liquid chromatography. The considered veterinary drugs were all detected at a level below their corresponding MRLs. Nanoflow liquid chromatographic methods gradients are usually longer than those of (ultra) high-performance liquid chromatography. However, the use of this approach and the complete elimination of matrix effects makes the use of matrix-matched calibration or the standard addition method unnecessary, a valuable feature considering the potential savings resulting from its implementation in laboratories. Other recent studies [50,51,52] demonstrate the potential of HRMS for the detection and quantification, in routine analysis, of a wide range of veterinary drug residues in multiple matrices.

The studies presented in this section and summarized in Table 2 are based on the detection of residues of known drugs. However, new drugs and new methods of application are being developed to overcome the detection of fraudulent practices. In this context, Dervilly-Pinel and co-workers [53,54] developed and validated a method to screen for β-agonists in bovines, based on a pure metabolomics approach. The method combined three biomarkers and bioinformatics to formulate a discriminant function to predict β-agonist treatment in bovines in an inexpensive, accurate, feasible, high-throughput test. Despite efforts to identify these three biomarkers, two of them remained unresolved. The untargeted workflow was, therefore, accredited to implement the innovative screening tool, demonstrating the power of HRMS in the analysis of veterinary drugs.

## 4. Analysis of Natural Toxins

By contrast with pesticides and veterinary drugs, toxins are not man-made. They are metabolites produced by living organisms that are typically not harmful to the organisms themselves but can adversely affect human or animal health when consumed [55]. Natural toxins have multiple sources, including plants (phytotoxins or plant toxins), fungi (mycotoxins), algae (phycotoxins or biotoxins) and bacteria (bacterial toxins). The diversity of these biological systems and the disparate properties of toxins present challenges to analytical chemists and wide-ranging food safety implications. Moreover, the chemical structures of the toxins can be altered by the metabolism of the organisms as part of their defence against xenobiotics, increasing the wide spectrum of possible occurring contaminants [56]. For the most prevalent and potent toxins in both animal feed and human food, regulatory limits have been set in European Union legislation [3,57,58,59].

To ensure food and feed safety and compliance with European legislation, several detection and screening methods have been developed for the analysis of natural toxins. Methods based on chromatographic analysis coupled to fluorescence or ultraviolet (UV) detection are progressively replaced by chromatographic separation coupled to tandem mass spectrometric analysers, such as triple quadrupoles. This technique is, however, limited to targeted analysis, making necessary the use of an analytical standard, which is a critical issue for modified toxin analysis [56]. In recent years, thanks to technological advances and improved affordability, HRMS-based methods have been developed to circumvent this issue. These methods allow screening for and quantification of hundreds of parent toxins and associated metabolites in food and feed samples.

Mycotoxins are the most studied toxins, for which multiple HRMS-based methods have been proposed in recent years. Zachariasova and co-workers developed, for example, methods to analyse multiple mycotoxins in cereals [60] and beer [61]. Depending on the matrix, different strategies for mycotoxin quantification were suggested. The use of isotopically labelled surrogates and analytes in pure solvent standards for the construction of calibration curves provided better recovery and repeatability of results for cereals, whereas a matrix-matched calibration was preferred for beer. In both studies, the authors insisted on the importance of the resolving power of the instrument used for the analysis, improving the selectivity of the detection in the presence of abundant co-eluting matrix components. Another method was developed by Lattanzio and co-workers [62] for the simultaneous determination of multiple mycotoxins in wheat flour, barley flour and crisp bread. A critical comparison between the HRMS method and a validated method based on triple quadrupole mass spectrometry showed similar performance in terms of detection limits (in the range of 0.1 to 2.9 µg/kg with the HRMS method, adequate to assess mycotoxin contamination in cereal foods at regulatory levels), recoveries, repeatability and matrix effects. In their conclusion, the authors described HRMS-based methods as a reliable and robust alternative tool for the routine analysis of major mycotoxins in foods, with the additional advantage of the possibility to perform retrospective analysis, and to search for mycotoxin metabolites. Ates and co-workers [63] developed a method combining an automated on-line sample clean-up and LC-HRMS for the analysis of multiple mycotoxins in maize, wheat and animal feed. With this approach, interfering matrix compounds with different chemical properties and macromolecules such as fats and proteins can be removed. The method was validated with good repeatability, and quantification limits were all acceptable with respect to legislative limits. Certified reference materials, which have been analysed as representative samples of maize, wheat and animal feed for the target compounds, demonstrated a high method accuracy. Moreover, the developed method was successfully employed in proficiency testing of animal feed samples, confirming its applicability for routine analysis. The same approach was utilized by authors to screen plant and fungal metabolites in wheat, maize and animal feed, using an empirical database of over 600 metabolites [64]. The wide applicability of the method was first demonstrated by the validation of 15 fungal and plant metabolites in maize, wheat and animal feed samples. The method was then applied to market samples. In addition to regulated and known secondary metabolites, 3 other mycotoxin metabolites were identified for the first time, demonstrating the capacity of HRMS full-scan analysis. The applicability of HRMS-based methods, in routine analysis, for the determination of multiple mycotoxins in complex matrices was also demonstrated in several studies [65,66,67].

Besides mycotoxins, biotoxins also pose a significant food safety risk, for which dedicated HRMS methods were proposed. Blay and co-workers [68] developed, for instance, an LC-MS platform for the non-targeted screening of two major classes (hydrophilic and lipophilic) of biotoxins commonly found in shellfish. Although two different modes of separation were employed for the two classes, authors insisted on the minimum required MS method development time and the possibility to easily extend the approach to other toxins or toxin analogues. Rúbies and co-workers [69] developed a high-throughput confirmatory quantitative method for the analysis of regulated biotoxins in fresh and canned bivalves. Thanks to the high resolving power of the Q-Orbitrap instrument, accurate mass data were obtained for each analyte—both the molecular ion and the selected fragment. The different compounds were, therefore, identified with high confidence and the risk of false positive results is limited. Using a matrix-matched calibration curve and a HRMS/MS acquisition mode, the method provided reliable quantitative results at the regulated concentration levels. This method is currently used in a routine laboratory in Spain. Finally, HRMS based methods were also developed for the analysis of phytotoxins, such as tropane alkaloids in animal feed [70] or teas [71] or alkenylbenzenes in pepper [72]. These studies, summarized in Table 3, demonstrate, once again, the versatility of HRMS and its suitability for routine analysis.

## 5. Multi-Class Analysis

The previous sections concerned HRMS analysis methods dedicated to the determination of a single class of contaminants, including pesticides, veterinary residues and toxins. However, in full-scan mode, high-resolution instruments are able to screen for a theoretically unlimited number of compounds in a single analysis. The simultaneous analysis of multi-class compounds is, therefore, possible with the HRMS approach. Food and feed can, indeed, be contaminated with different types of undesired compounds. Pesticides and mycotoxins can, for example, be found in crops, just as pesticides and veterinary drug residues can be found in milk. Sample preparation is not addressed in this review but is of the utmost importance in non-targeted methods and is even more challenging in this context, with the high variability of physicochemical properties of the screened compounds. The impact that sample preparation and instrument parameters on data quality and chemical coverage is illustrated in the study of Knolhoff and co-workers [73].

In a feasibility study, Pérez-Ortega and co-workers [74] demonstrated that a time-of-flight instrument was suitable for the screening of 625 multiclass food contaminants in baby-food samples (containing meat and vegetables). Due to the resolving power of the instrument, retention time, isotopic profile and fragment ions, almost all the components, including pesticides, veterinary drugs, mycotoxins and other contaminants of concern, were successfully resolved. Only three pairs of compounds, out of the 76% of component involved in isobaric groups, were not resolved using these tools. Highlighted weaknesses of this approach were chromatographic issues with highly polar species, low sensitivities for selected compounds, which does not map well against electrospray ionization, and quantitation of compounds affected by signal suppression effects due to co-eluting matrix components or analytes. However, these issues are not specific to the HRMS approach and also affect the traditional triple quadrupole-based targeted approach. In contrast, only the HRMS approach is capable of screening for such a large number of contaminants. Residue analysis of infant food are is of high importance because this specific age group is more sensitive to several chemicals due to their high food intake/body weight ratio and the immaturity of their defence systems against chemical stressors [75]. A specific directive was, therefore, defined by the European Commission for such foodstuffs [76]. It prohibits the use of certain very toxic pesticides in production and requires that infant formula and follow-on formula contain no detectable levels of pesticide residues (meaning <0.01 mg/kg). The detection of residues at such a challengingly low level was partially met in another study from Gómez-Pérez and co-workers [77]. More than 300 pesticide and veterinary drug residues were quantified with a validated method presenting limits of detection from 0.5 to 50 mg/kg and limits of quantification between 10 and 100 mg/kg.

A study by Cotton and co-workers [78] provides a relevant example of the potential of HRMS for the analysis of water. They developed and validated (repeatability, selectivity, linearity and matrix effect) a multi-residue targeted method for the analysis of more than 500 pesticides and drugs in water. More than 30 different compounds were detected in 20 tap water samples collected in and around Paris but at level lower than 0.1 µg/L, the European Union limit for pesticides in drinking water. In another study, Albergamo and co-workers [79] developed an HRMS-based method for the identification and quantification of 33 polar micropollutants, representative for several classes of emerging contaminants (herbicides, sweeteners, pharmaceutically active compounds, anticorrosive agents and industrial chemicals), in natural drinking water sources. The authors argued that most polar micropollutants are overlooked by the current regulatory actions, resulting in the need to use accurate, sensitive and robust analytical tools to efficiently monitor source waters. The method detection limit for the 33 considered micropollutants in riverbank filtrate water ranged between 8 and 83 ng/L and was lower than 20 ng/L for 27 of them. The authors also highlighted that the developed method is suitable for a larger number of compounds, such as analogues and metabolites of the micropollutants, and the database of target compounds is extendable and could be used for retrospective suspect screening.

However, the use of routine non-target analysis is still uncommon for most environmental monitoring agencies and environmental scientists [80]. Compound identification in targeted and suspect-screening analysis (using a non-targeted data-acquisition approach) rely on reference standards, databases containing compounds structure, isotope pattern, presence of additional adducts, chromatographic retention behaviour, fragmentation information, and other experimental evidence. Purely non-targeted analysis aims at identifying compounds present in the sample without prior information. Data processing in such an approach is laborious, and the achievement of quantitative results remains difficult. These statements are well illustrated in a study by Schymanski and co-workers, who reviewed a collaborative trial on water analysis using non-targeted screening with HRMS. It was observed that the analytical methods are already reasonably well harmonised, contrary to processing workflow and used databases. Targets from some laboratories were found to be suspects or unknowns in other laboratories. Participants in the trial expressed the need to harmonise information sources. Authors insisted on the fact that enhancing this by uploading mass spectra of target compounds to an open access database would help improve the success of target, suspect and even non-target screening immensely. However, it is likely that several degradation products of industrial contaminants and pharmaceuticals would not be found in any current libraries. Improvement of the entire data treatment process is necessary to more extensively characterize water samples for a non-targeted approach and answer questions regarding the origin of the contamination or the dynamics of the contaminants [80].

Several other HRMS-based methods were developed for multi-class residue analysis, such as pesticides, veterinary drugs and mycotoxins, and quantitative determination in both bakery raw materials and finished products [81]. The authors highlighted that a more satisfactory performance may still be seen by way of a robust triple quadrupole MRM analysis. HRMS-generated data, however, allow for additional flexibility in post-acquisition processing, with the consequent advantage of the possibility to execute retrospective data mining. Several recent studies were identified for the determination of various undesired residues in multiple matrices, including feed, honey, vegetables, cereals, tea, botanical nutraceuticals or edible insects, with a growing interest in alternatives to the increasing food demand [82,83,84,85,86,87,88]. These numerous studies, summarized in Table 4, confirmed the suitability of HRMS for multi-class residue analysis, in particular, with hundreds of compounds screened in a single run, which was possible due to full-scan mode.

## 6. Food Authenticity

Besides analysis aiming to detect contaminants and residues in food and feed, HRMS has been used to assess food authenticity and detect fraud or adulteration. Food fraud is motivated by economic gain but can represent a serious health risk for consumers. By assessing food authenticity, consumers are protected from purchasing products of inferior quality or with incorrect descriptions and honest traders are defended from unfair competition. Wine, spirits, olive oil, fish, meat, cheese, honey and herbs and spices represent the most commonly reported adulterated foods [89].

HRMS analysis with untargeted data acquisition and chemometrics tools, such as principal component analysis, are a powerful combination for the evaluation of food authenticity. Among the wide range of analysed compounds, chemometric tools can decrease the number of detected features remarkably and suggest characteristic markers responsible for different types of authenticity issues, such as adulteration, variety or geographical origin discrimination, organoleptic profiles, ripening and method production.

Rubert and co-workers used metabolic fingerprinting for wine authentication according to the grape varieties. The validated discriminant analysis models based on the acquired data were able to correctly classify 95% of over 300 wine samples. Using online libraries, the markers used for wine authentication were tentatively identified and corresponded to different flavanol glucosides and polyphenols. Hrbek and co-workers [90] used a similar approach to identify garlic origin, which is known to influence its organoleptic properties. The data generated by an HPLC-HRMS analysis of 47 samples of garlic from different geographical origins were used to construct statistical models to authenticate garlic origin. A number of robust targets, including free amino acids and characteristic sulphur compounds, were identified as the most suitable markers. The last selected example was a study by Fiorino and co-workers [91], aiming to assess fish authenticity and discriminate between wild-type and farmed salmon. Wild salmon is, indeed, known to be richer in the more valuable omega-3 fatty acids. A fast HRMS analysis method, using an Orbitrap instrument, was developed and combined with data integration via principal component analysis. The analysis of wild-type and farmed salmon samples from different origins led to the conclusion that saturated and polyunsaturated fatty acids with 20 or 22 carbon atoms on their side chains were the most suitable markers to discriminate between the two types of salmons. The developed method was successfully applied to commercial samples.

These few examples demonstrate the power of HRMS analysis combined with chemometrics tools in food authenticity assessment. Identification is based on specific markers from the theoretically unlimited number of components detected during the analysis. Chemometrics tools enable the highlighting of these specific markers among the massive amount of data. If present in libraries, these specific markers can be identified to achieve a more robust authentication method. Due to the great amount of time and number of reference samples required to develop a model, this approach is currently limited to a few food products, mostly with high added value. However, associations such as the Association of Official Analytical Collaboration (AOAC) International are currently working on guidelines for method development and validation, including untargeted approaches [92].

## 7. Conclusions

In recent years, food and feed analysis based on HRMS coupled to chromatographic separation techniques has become increasingly common, and today, some developed methods are used in routine analysis. Contrary to targeted methods using low-resolution instruments, such as highly popular triple quadrupole instruments, HRMS-based methods using untargeted data acquisition are able to record a theoretically unlimited number of compounds in full-scan mode. The development of HRMS acquisition methods is relatively simple compared to the time-consuming and standard-requiring development and optimisation of MRM methods. Moreover, due to untargeted data acquisition, retrospective analysis is possible and could be relevant when new contaminants or residues are discovered.

Triple quadrupole instruments operating in MRM mode generally demonstrate higher sensitivities than HRMS instruments operating in full-scan mode. However, using the quadrupole of a Q-TOF or Q-Orbitrap to isolate a narrower mass range frequently improves HRMS sensitivity [15]. Moreover, components producing many fragments grant superior HRMS sensitivity since the compounds are preferably detected as unfragmented precursor ions. The sensitivity gap between the two technologies has likely narrowed over the last decade, and this process will probably continue. Sensitivity and quantification issues due to the poor ionization of certain targets and coeluting matrix components can also be encountered. However, these issues are not specific to HRMS-based methods and are also encountered with a triple quadrupole instrument. Nevertheless, the numerous studies and developed methods presented in this review demonstrate the capability of HRMS to detect several types of components at levels in compliance with the current relevant legislation. Moreover, alternative and innovative approaches using HRMS have recently been developed, such as untargeted metabolomics, allowing screening for banned compounds.

Currently, HRMS-based analysis is limited to components characterized in databases. Purely non-targeted screening without any prior information on the compounds remains challenging, and efforts towards developing computational tools are necessary to enable the use of this approach in the future. Therefore, the current priority is the expanding and the disseminating of libraries and databases for a wide range of contaminants, residues and associated transformation products. This will extend the scope of HRMS analysis for food and feed samples and make this approach essential.

## Figures and Tables

**Table 1 foods-10-00601-t001:** Selected studies on pesticides analysis by high-resolution mass spectrometry (HRMS).

Instrument and Scanning Technique	Matrix	Number of Analytes	Method Sensitivity	Reference
HPLC-ESI-Q-ToF with full-scan MS suspect screening in the first injection and target MS/MS confirmation in the second injection	Cucumber and orange	317	48.9% of the analytes detected and 17.3% confirmed at 1 µg/kg 83.9% of the analytes detected and 77.6% confirmed at 10 µg/kg 98.1% of the analytes detected and 83.9% confirmed at 50 µg/kg	[27]
UHPLC-ESI-Q-Orbitrap with full-scan MS for screening and quantification in the first injection and target MS/MS confirmation in the second injection	Apple, banana, grape, orange, strawberry, carrot, potato, tomato, cucumber, and lettuce.	166	87.3–92.7% of the analytes with LOD and LOQ ≤ 5 μg/kg Most of the analytes with LOQ ≤ 10 µg/kg	[28]
UHPLC-ESI-Q-Orbitrap with full-scan MS for screening and quantification in the first injection and target MS/MS confirmation in the second injection	Apple, banana, grape, orange, strawberry, carrot, potato, tomato, cucumber, and lettuce.	451	85% of the analytes with LOD and LOQ ≤ 5 μg/kg Most of the analytes with LOQ ≤ 10 µg/kg	[29]
UHPLC-ESI-Q-Orbitrap with simultaneous full-scan MS and single MS/MS scan	Tomato, pepper, orange and green tea	139	>90% of the analytes with LOD and LOQ ≤ 10 μg/kg	[30]
HPLC-ESI-Q-Orbitrap with simultaneous full-scan MS and MS/MS scans	Swiss groundwater samples	519 target analytes and 1256 suspect analytes	78% of the target analytes with LOQ ≤ 10 ng/L	[31]
Ion Chromatography ESI-Q-Orbitrap with simultaneous full-scan MS and MS/MS scans	Grapes, wheat and honey	11	7 analytes with LOQ ≤ 10 µg/kg, 9 analytes with LOQ ≤ 50 µg/kg, 11 analytes with LOQ ≤ 100 µg/kg	[32]
GC-Negative Chemical Ionization- Q-ToF with simultaneous full-scan MS and MS/MS scans	Tomato	70	76% of the analytes with LOD ≤ 1 µg/kg 57% of the analytes with LOQ ≤ 1 µg/kg 99% of the analytes with LOD ≤ 5 µg/kg 93% of the analytes with LOQ ≤ 5 µg/kg	[34]
GC-Electron Ionization- Q-Orbitrap with simultaneous full-scan MS and MS/MS scans	Pear, banana, watermelon and strawberry	191	LOD at 1 µg/kg in pear, banana, watermelon and 5 µg/kg in strawberry. LOQ at 5 µg/kg in all 4 matrices	[35]

LOD, limits of detection LOQ, limits of quantification.

**Table 2 foods-10-00601-t002:** Selected studies on veterinary drug residues analysis by HRMS.

Instrument and Scanning Technique	Matrix	Number of Analytes	Method Sensitivity	Reference
UHPLC-ESI-Orbitrap with full-scan MS	Bovine urine	87	33.3% of the analytes with LOD ≤ 1 µg/L 98.9% of the analytes with LOD ≤ 10 µg/L	[45]
UHPLC-ESI-ToF with full-scan MS	Honey	200	75.5% of the analytes with LOD ≤ 20 µg/kg 89.3% of the analytes with LOD ≤ 50 µg/L	[46]
UHPLC-ESI-Q-ToF with full-scan MS suspect screening in the first injection and target MS/MS confirmation in the second injection	Bovine, rabbit, poultry, goat and pork feeds	116	40% of the analytes detected and 10% confirmed at 0.02 mg/kg in all 5 matrices 75% of the analytes detected and 55% confirmed at 0.2 mg/kg in all 5 matrices	[47]
HPLC-ESI-Orbitrap with full-scan MS	Pig, poultry, cattle lamb and fish feed	48	50% of the analytes with LOD ≤ 10 µg/kg 94% of the analytes with LOD ≤ 20 µg/kg 79% of the analytes with LOQ ≤ 50 µg/kg 98% of the analytes with LOQ ≤ 100 µg/kg	[48]
Nanoflow LC-ESI-Q-Orbitrap with simultaneous full-scan MS and MS/MS scans	Honey, veal muscle, egg and milk	87	38% of the analytes with LOQ ≤ 0.1 µg/kg in all 4 matrices 100% of the analytes with LOQ ≤ 1 µg/kg in all 4 matrices	[49]
UHPLC-ESI-Q-Orbitrap with simultaneous full-scan MS and MS/MS scans	Pork meat	37	LOD between 0.8 and 3.5 µg/kg LOQ between 2.4 and 10.5 µg/kg	[50]
UHPLC-ESI-Q-Orbitrap with full-scan MS	Bovine, chicken and porcine meat	164	10.9% of the analytes confirmed at 1 µg/kg 32.3% of the analytes confirmed at 10 µg/kg 83.5% of the analytes confirmed at 100 µg/kg	[51]
UHPLC-ESI-Q-Orbitrap with simultaneous full-scan MS and MS/MS scans	Milk	105	58% of the analytes detected and 51% confirmed at 1 µg/kg 96% of the analytes detected and 84% confirmed at 10 µg/kg	[52]

**Table 3 foods-10-00601-t003:** Selected studies on natural toxins analysis by HRMS.

Instrument and Scanning Technique	Matrix	Number of Analytes	Method Sensitivity	Reference
UHPLC-ESI-Orbitrap and UHPLC-ESI-ToF with full-scan MS	Maize, barley and wheat	11	5 analytes with LOD ≤ 10 µg/kg and with ToF 2 analytes with LOD ≤ 10 µg/kg with Orbitrap 9 analytes with LOD ≤ 25 µg/kg with ToF 9 analytes with LOD ≤ 25 µg/kg with Orbitrap 5 analytes with LOQ ≤ 25 µg/kg with ToF 3 analytes with LOQ ≤ 25 µg/kg with Orbitrap 9 analytes with LOQ ≤ 50 µg/kg with ToF 10 analytes with LOQ ≤ 50 µg/kg with Orbitrap	[60]
UHPLC-Atmospheric pressure chemical ionisation-Orbitrap with full-scan MS	Pale lager, non-alcoholic, and black lager beers	32	16% of the analytes with the lowest calibration level ≤2 µg/L 87% of the analytes with the lowest calibration level ≤10 µg/L	[61]
HPLC-ESI- High-energy collision dissociation–Orbitrap with full-scan MS	Wheat flour, barley flour, wheat crisp bread and rye crisp bread	9	Analytes detected in 0.1–1.6 µg/kg range and confirmed in 0.1–3.4 µg/kg range in all 4 matrices	[62]
HPLC-ESI-High-energy collision dissociation–Orbitrap with full-scan MS	Maize, wheat and animal feed	6	LOD between 2 and 150 µg/kg in all 3 matrices LOQ between 5 and 375 µg/kg in all 3 matrices	[63]
HPLC-ESI-High-energy collision dissociation–Orbitrap with full-scan MS	Maize, wheat and animal feed	15	99% identification rate in multiple replicates at 250 µg/kg in all 3 matrices	[64]
HPLC-Atmospheric pressure chemical ionisation-Q-Orbitrap with simultaneous full-scan MS and MS/MS scans	Forage maize and maize silage	8	Analytes detected in 11–88 µg/kg range and confirmed in 20–141 µg/kg range in both matrices	[65]
UHPLC-ESI-TOF with full-scan MS	Maize	9	5 analytes with LOD ≤ 1 µg/kg 7 analytes with LOD ≤ 25 µg/kg 5 analytes with LOQ ≤ 2 µg/kg 7 analytes with LOQ ≤ 50 µg/kg	[66]
UHPLC-ESI-Q-Orbitrap with simultaneous full-scan MS and MS/MS scans	Corn, rice, wheat, almond, peanut and pistachio	26	46% of the analytes with LOD ≤ 0.1 µg/kg 76% of the analytes with LOD ≤ 1 µg/kg 54% of the analytes with LOQ ≤ 0.5 µg/kg 81% of the analytes with LOQ ≤ 5 µg/kg	[67]
HPLC-ESI-Orbitrap with full-scan MS and separated LC methods for lipophilic and hydrophilic toxins	Mussel tissue	10 lipophilic12 hydrophilic	LOD between 0.041 and 5.1 µg/L for lipophilic toxins LOD between 3.4 and 14 µg/L for hydrophilic toxins	[68]
UHPLC-ESI-Q-Orbitrap with simultaneous full-scan MS and MS/MS scans	Fresh and canned bivalves	10	LOQ at 25 µg/kg	[69]
HPLC-ESI-High-energy collision dissociation–Orbitrap with full-scan MS	Chicken feed	12	LOQ between 5 and 25 µg/kg	[70]
HPLC-ESI-High-energy collision dissociation–Orbitrap with full-scan MS	Tea and herbal teas	13	LOQ between 5 and 20 µg/kg	[71]
GC–Electron Ionization-Q-Orbitrap with full-scan MS	Pepper	8	LOD between 0.01 and 0.02 mg/kg LOQ at 0.2 mg/kg	[72]

**Table 4 foods-10-00601-t004:** Selected studies on HRMS analysis of multi-class contaminants.

Instrument and Scanning Technique	Matrix	Number of Analytes	Classes of Analytes	Method Sensitivity	Reference
UHPLC-ESI-Collision induced dissociation-Q-ToF with full-scan MS	Orange, tomato and baby food	625	Pesticides, veterinary drugs, mycotoxins, food-packaging contaminants, perfluoroalkyl substances or nitrosamines	80–85% of the analytes detected at 50 µg/kg	[74]
UHPLC-ESI-Q-Orbitrap with simultaneous full-scan MS and MS/MS scans	Water	539	Pesticides and drugs	44% of the analytes with LOD ≤ 0.001 µg/L 84% of the analytes with LOD ≤ 0.01 µg/L 99% of the analytes with LOD ≤ 0.1 µg/L	[78]
UHPLC-ESI-Collision induced dissociation-Q-TOF with full-scan MS	Surface water and groundwater	33	Herbicides, sweeteners, drugs, anticorrosive agents and chemicals	LOD between 0.009 and 0.093 µg/L	[79]
UHPLC-ESI-Orbitrap with full-scan MS	Milk, flours and minicakes	36	Pesticides, antibiotics and mycotoxins	17% of the analytes with LOD ≤ 1 µg/L 72% of the analytes with LOD ≤ 10 µg/L 83% of the analytes with LOD ≤ 100 µg/L	[81]
UHPLC-ESI-Q-Orbitrap with simultaneous full-scan MS and MS/MS scans	Botanical Nutraceuticals	16	Pesticides and mycotoxins	LOQ between 0.2 and 6.25 µg/kg	[82]
UHPLC-ESI-Q-Orbitrap with full-scan MS	Edible insects	77	Pesticides, (veterinary) drugs and mycotoxins	75 analytes detected in 1–100 µg/kg range	[83]
UHPLC-ESI-ToF with full-scan MS	Tea brew and tea leaves	32	Pesticides, mycotoxins, process-induced toxicants and packaging contaminants	81% of the analytes detected at 10 µg/kg 63% of the analytes quantified at 10 µg/kg	[84]
HPLC-ESI-High-energy collision dissociation–Orbitrap with full-scan MS	Nutraceutical products (green tea and royal jelly)	260	Pesticides and mycotoxins	LOD between 0.5 and 10 µg/kg LOQ between 1 and 20 µg/kg	[85]
UHPLC-ESI-High-energy collision dissociation–Orbitrap with full-scan MS	Cattle feed	77	Veterinary drugs, ergot alkaloids, plant toxins and other undesirable substances	52% of the analytes with LOQ ≤ 5 µg/kg 87% of the analytes with LOQ ≤ 25 µg/kg	[86]
HPLC-ESI-Q-Orbitrap with simultaneous full-scan MS and MS/MS scans	Leek, wheat, and tea	389	Pesticide, mycotoxins, and pyrrolizidine alkaloids	82% of the analytes with LOQ ≤10 µg/kg in leek 81% of the analytes with LOQ ≤10 µg/kg in wheat 61% of the analytes with LOQ ≤10 µg/kg in tea	[87]
HPLC-ESI-High-energy collision dissociation–Orbitrap with full-scan MS	Honey	350	Pesticides and veterinary drugs	95% of the analytes with LOD ≤ 10 µg/kg	[88]

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
