# Peer review of "Suitability of High-Resolution Mass Spectrometry for Routine Analysis of Small Molecules in Food, Feed and Water for Safety and Authenticity Purposes: A Review"

_foods, 2021, doi:10.3390/foods10030601_

Round 1
Reviewer 1 Report
The manuscript has been significantly improved according to the reviewers' comments and need a minor revision.
Minor revision:
Table 1 and Table 4 are both related to pesticides. What are the differences between them? Could they be merged into a single table?
Otherwise, the table captions should describe the differences among them with a more detailed explanation of both tables.
Author Response
The caption of Table 4 was wrong and has been modified as "Selected studies on HRMS analysis of multi-class contaminants". The caption of Table 4 is now describing its content.
Reviewer 2 Report
This is a resubmitted review on the small molecule analyses in food, feed, and water by high-resolution mass spectrometry. Many points of concern from the previous reviewers have been noted. Here are some additional questions that need to be addressed.
Table 4. accompanied Section 5. Multi-class analysis. Since this section was supposed to report the "multi-class residue analysis (including pesticides, veterinary residues, and toxins)," should the title of the table include small molecules other than pesticides? In this vein, should the content of Table 4 also reflect which or how many different classes of small molecules being detected in the referred references?
Section 6 focused on food authenticity, which was achieved through "metabolic fingerprinting (detecting) flavonol glucosides and polyphenols, …. amino acids, and….fatty acids". If this review's title was on "pesticide, veterinary drug residues, and natural toxins," section 6 was unrelated and should be deleted. Otherwise, the title needs to be more general to include the small molecule detections in food authenticity.
Author Response
The caption of Table 4 was adapted to “Selected studies on HRMS analysis of multi-class contaminants” to more accurately describe its content. As suggested, a column was added to this table to indicate to the reader, the classes of analytes covered in each study.
The major part of the review covered the analysis of small molecules including pesticides, veterinary drugs and natural toxins. However, to demonstrate the versatility of high resolution mass spectrometry, we wanted to conserve the section dedicated to food authenticity. We therefore adapted the title of the review to “Suitability of high-resolution mass spectrometry for routine analysis of small molecules in food, feed and water for safety and authenticity purposes: A review”.
Reviewer 3 Report
The presented work is interesting and well planned. Key aspects were covered: pesticides, veterinary drug residues, natural toxins in food, feed and water .
- (246) Could you detail which pharmakokinetics parameters are important to establish a withdrawal period.
- (eg. 509)The abbreviation HRMS should be used in all document.
- Table 4 - 0.25-6.25 ng/g - in all document you use μg/kg or μg/L so I propose to to change the unit ng/g
- (233) The topic of potential side effect of redidues of drugs in animal products should be develop. The main groups of veterinary drugs shoul be given in article.
- (276) ppm should be change to μg/kg
Author Response
- (246) The parameters involved in withdrawal period establishment were briefly described and a reference to the review of Lees et al. was added. The sentence “This withdrawal period is drug related with specific absorption and elimination rates and also depends on the route of administration and the dosage regimen[40].” was added to the manuscript.
- Except for their first occurrence and in the abstract, the terms “high-resolution mass spectrometry” were replaced by its abbreviation “HRMS” throughout the manuscript.
- The unit ng/g in Table 4 was replaced by µg/kg for consistency with the other values presented in that table but also in the rest of the manuscript.
- We considered that the effects of veterinary drugs residues on human health was outside of the scope of this analytical manuscript was not further developed. Interested readers can have a look at the suggested reference, i.e. the review of Baynes et al. entitled “Health Concerns and Management of Select Veterinary Drug Residues”.
The major classes of veterinary drugs including antibiotics, antiprotozoals, anthelmintics, anti-inflammatory, corticosteroids and hormones substitutes were added to the manuscript in the first sentence of section 3 dedicated to the analysis of veterinary drug residues. - In the context of line 276, “ppm” was used to express a tolerated mass deviation for compounds identification, a parameter in HRMS data processing. The “ppm”, here, did not refer to the expression of a concentration and we therefore preferred to keep “ppm” in the manuscript in this context.
This manuscript is a resubmission of an earlier submission. The following is a list of the peer review reports and author responses from that submission.
Round 1
Reviewer 1 Report
This review discussed the employment of high-resolution mass spectrometry (HRMS) in the routine analysis of small molecules in food, feed, and water. However, this review needs major structural changes for a better flow, and figures and tables to guide the readers.
First of all, the discussion between low-resolution MS vs HRMS, GC vs. LC, scanning techniques (MRM vs. full scan), targeted vs. untargeted scanning, etc. should come before any applications. Such section should be accompanied by figures for a better illustration. Without such section and figures, it will be very difficult for the readers who have no solid background in MS to understand the necessity of using HRMS. This review also seems to list papers using low-resolution MS or HRMS without clear logic. In addition, since the common analytes were detected in food, feed, and water, it was not clear why water was separately discussed in Section 6.
A Table of Content should be included for better navigation.
A table should also be included to summarize the papers referred to in this review. Information such as scanning techniques, LOD/LOQ, sample sources, etc. should be included in this table.
The paragraph starting from line 193 seems out of place. It referred to a “masterpiece” using triple quadrupole MS. No explanation was given as to why this particular study was a “masterpiece,” and only future tense was used to “predict” that “HRMS will become a key element in the analysis of pesticides” when HRMS mentioned in the paragraphs above had successfully detected 317 (line 130), 166 and 451 (line 139), 139 pesticides (line 150).
Reviewer 2 Report
This review deals with the recent applications of high-resolution mass spectrometry in different sectors. The topic of the manuscript is of interest to the scientific community, however, this review is too general since it describes different fields of application. Moreover, this type of review requires some information in table format to be more readable.
Some comments/suggestions:
- According to the title, the different sections should be renamed…
- section 2 pesticides: they are present in food and feed as well as in water (new section)… Should you divide it into food and feed (pesticides, toxins, and veterinary drugs) and then water? Then, after the water section, the authors included food authenticity, maybe should be after food or within the food...Then, The multiclass analysis, this approach could be applied to water? Maybe the water analysis should not be considered a separated section, just a suggestion…
- Lines 11-15: this sentence should be split, is too long...
- Lines 23-29: the authors should include the reference to this paragraph.
- Lines 56-67: Should not be a new paragraph, is following the previous idea. The authors should include also the references required.
- Lines 98-104: Please add the reference.
- Section 2 and may be extended to the other sections: some data should be presented in tables with common data to be comparable between methods approach, samples, limits of identification,...Furthermore will be easier to read...
- Lines 208-210: Should be indicated in the text the regulation number
- Lines 225-234: Add reference.
- Line 258: "In more recent methods, the number of screened drugs increased,...". Do you mean recent studies?
- Lines 289-291: If the authors present these results in a table it will be easier for the reader, otherwise this sentence says very little...
- Lines 318-320: Add a reference
Reviewer 3 Report
I am a bit confused by this review. On one hand the paper is generally well written, and it does have some useful basic information in it. However, the main stated point of the review – to show that high-resolution mass spectrometry could be used for the routine analysis of food, feed and water is moot. High-resolution mass spectrometry is already used for routine analysis of food, feed and water I know, I do it regularly and it is even stated in this paper that people use high resolution mass spec for this sort of analysis. There is therefore no need to prove how useful the technique is, we already know it is useful. Given this – what does the paper add?
The point about Article 152 of the Treaty establishing the European Community is technically true but what about the rest of the world. The USA for example or Japan or Australia.
It is repeatedly stated that the number of compounds that can potentially be screened by high resolutions mass spectrometry is almost unlimited. This is not true. There is a limit to the separation space available using any analytical technique. Any instrument can only scan so fast and because the possible targets vary by orders of magnitude in size, polarity, mass etc. there is no method that can detect everything that might be present.
It is true that Maximum levels have been set for certain contaminants (e.g., mycotoxins, dioxins, heavy metals, nitrates and chloropropanols) but high resolution mass spectrometry is already used to measure them. Again what is the point here?
Aside from the fact that we already know that high-resolution mass spectrometry could be used for the routine analysis of food, feed and water the authors don’t present any new information. The paper is entirely text, huge wall of it. The authors just give very brief summaries of a few papers but they don’t synthesise the information and draw out commonalities or new information that might be useful to the reader. What is the point of just saying author A found this and author B found that then author C found the other. It does not add anything to present information this way. The are no tables showing what techniques were used for which compounds or a timeline showing how detection limits have gotten better over time for example or which countries are using what methods?
The authors correctly state that Sample preparation is of the utmost importance (section 5) but then they don’t discuss it at all? Sample prepearion is the most important part of an analysis. How can you ignore it?
In section 8 the authors state that “alternative and innovative approaches using HRMS have recently been developed, such as untargeted metabolomics, allowing screening for banned compounds. No!, this is totally wrong. Metabolomics as a field looks at natural metabolites it is not used to screen for banned compounds. It is not even used in regulatory toxicology.
The authors also state in section 8 that HRMS-based analysis is limited to components characterized in databases. This is not really true. While databases help HR techniques such as Q-TOF mass spec does help identify unknown compounds.